# Freestanding non-covalent thin films of the propeller-shaped polycyclic aromatic hydrocarbon decacyclene

Alex van der Ham[1,4], Xue Liu [1,2,4], Dario Calvani [1], Adéla Melcrová [3], Melania Kozdra [1], Francesco Buda[1], Herman S. Overkleeft[1], Wouter H. Roos [3], Dmitri V. Filippov [1] & Grégory F. Schneider [1✉]

Molecularly thin, nanoporous thin films are of paramount importance in material sciences. Their use in a wide range of applications requires control over their chemical functionalities, which is difficult to achieve using current production methods. Here, the small polycyclic aromatic hydrocarbon decacyclene is used to form molecular thin films, without requiring covalent crosslinking of any kind. The 2.5 nm thin films are mechanically stable, able to be free-standing over micrometer distances, held together solely by supramolecular interactions. Using a combination of computational chemistry and microscopic imaging techniques, thin films are studied on both a molecular and microscopic scale. Their mechanical strength is quantified using AFM nanoindentation, showing their capability of withstanding a point load of 26 ± 9 nN, when freely spanning over a 1 μm aperture, with a corresponding Young's modulus of 6 ± 4 GPa. Our thin films constitute free-standing, non-covalent thin films based on a small PAH.

[1] Leiden Institute of Chemistry, Leiden University, Einsteinweg 55, 2333 CC Leiden, The Netherlands. [2] State Key Laboratory for Mechanical Behavior of Materials, Xi'an Jiaotong University, 710049 Xi'an, China. [3] Zernike Institute for Advanced Materials, Rijksuniversiteit Groningen, Nijenborgh 4, 9747 AG Groningen, The Netherlands. [4] These authors contributed equally: Alex van der Ham, Xue Liu. ✉email: g.f.schneider@chem.leidenuniv.nl

Nanometer thin, porous thin films hold promise for a wide range of applications, ranging from hydrogen fuel cells and desalination, to biomedical applications and nanoelectronics[1–7]. The interest in nanometer thin films in particular stems from the fact that such thin films constitute the best trade-off between permeability and selectivity[8,9]. This is in part due to significantly diminished hydrodynamic resistance towards water molecules, as well as by being less prone to fouling and clogging of the pores in these thin films[10]. When designing novel thin films, control over the functionality of the film, in terms of chemical composition, thickness, pore size and pore charge is therefore crucial. One particularly successful approach to make such nanometer thin films is through molecular self-assembly (MSA)[11–13], which allows film properties to be tuned by incorporation of specific functional groups into the molecular building blocks used to make the film. However, a major hurdle in the bottom-up synthesis of nanometer thin films is the requirement to crosslink the constituent building blocks in order to achieve mechanical stability. This is typically achieved either via the incorporation of reactive functional groups into the monomeric building blocks, by thermal annealing[14,15], or photon-[16] or electron irradiation[17]. These techniques often lead to loss of chemical definition, resulting in poor sample-to-sample reproducibility. Bottom-up film formation, without the need for annealing but with good mechanical stability thus constitutes the ultimate goal. Within the field of reticular chemistry, several types of 2D non-covalent organic frameworks (non-COFs), like supramolecular organic frameworks (SOHs) and hydrogen-bonded organic frameworks (HOFs) are at the forefront of development in this direction[18–21]. Supramolecular frameworks based purely on π-π stacking interactions are rare, with only a single example in recent literature reporting on a 3D crystalline network[22]. Entirely 2D thin films based solely on non-covalent interactions between small molecules, like the one reported here, are not known in the literature to date.

Previously, Langmuir-Blodgett (LB) thin films composed of the small propellerene, decacyclene, have been developed for use in light-emitting devices[23,24]. The stacking behavior of this molecule and the mechanical properties of such films were, however, unexplored. This prompted us to undertake a combined study, initially using DFT computations in conjunction with Molecular Dynamics (MD) simulations, to get a molecular picture of the packing of decacyclene at the air-water interface, in order to investigate the potential use of decacyclene for MSA. Thin films were then synthesized and characterized using a range of high-resolution microscopic techniques (AFM, SEM and TEM) to get a

microscopic image of the films. We found the Van der Waals forces, which hold the molecules in these films together, are strong enough to allow them to be free-standing over micrometer distances. State-of-the-art AFM nanoindentation then allowed quantification of the mechanical strength of these unique thin films withstanding a point load of $26 \pm 9$ nN, with a Young's modulus of $6 \pm 4$ GPa. This is on the same order of magnitude as found for other 2D materials (e.g. MOFs and COFs)[25,26]. Our decacyclene thin films therefore constitute a PAH-based, non-covalent thin films, with a mechanical strength on par with that of known non-covalent membranes. These results therefore serve as an important step towards the fabrication of mechanically stable thin films, not requiring annealing of any kind.

## Results and discussion

**Langmuir–Blodgett experiments**. Decacyclene[27] films were prepared in a KSV minithrough, using Millipore® ultrapure water as subphase. Thin films were subjected to several compression-decompression cycles and the surface pressure monitored throughout (Fig. 1a, b). A strong hysteresis was observed, showing smaller mean molecular areas (MMA) with each cycle, going from an initial onset of 40 Å²/molecule to only 22 Å²/molecule[28]. This is a known phenomenon and indicates that the decacyclene monolayer is unstable at the air-water interface, and molecular rearrangements take place during the compression process. These molecular rearrangements entail the adaptation of a thermodynamically more favorable orientation of the molecules, which is known to be the direct result of factors such as intermolecular interactions, phase transitions and lateral diffusion[29].

**Molecular dynamics simulations**. To study the molecular processes taking place at the air-water interface during the compression, MD simulations were performed. The model system used consisted of a box of water molecules, with a total of 30 or 60 decacyclene molecules on each side (Fig. 2a, b). Input geometries for decacyclene were obtained from DFT computations at PBE/6-31 G(d,p). Optimized geometries with a $D_3$ symmetry were computed to be thermodynamically preferred over those with a $C_2$ symmetry ($\Delta G_{gas} = -0.3$ kcal mol$^{-1}$; Supplementary Fig. 10), in line with crystallographic data[30]. During the simulations, no interconversion from the $D_3$ conformer to the $C_2$ conformer is observed, whereas when simulations are performed starting with decacyclene in a $C_2$ conformation, irreversible transition to the $D_3$ conformer takes place during the pre-equilibration period. After pre-equilibration of the system at 300 K, the average

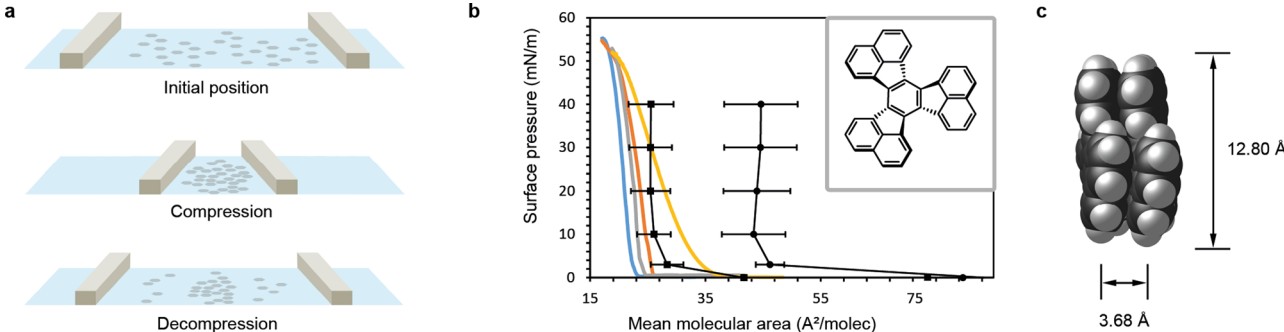

**Fig. 1 Langmuir–Blodgett experiments and thin film formation of decacyclene. a** Schematic representation of the compression–decompression of decacyclene molecules at the air–water interface in a Langmuir–Blodgett through. **b** Representative experimental Langmuir–Blodgett isotherms for two consecutive compression/decompression cycles on a single sample, showing surface pressure as a function of mean molecular area. Black lines are isotherms computed using MD simulations, for systems containing 30 (●) and 60 (■) molecules, respectively. Reported values are averaged over 3 separate MD simulations with error bars indicating standard deviations. Insert shows the molecular structure of decacyclene. **c** Optimized geometry of a decacyclene dimer computed at PBE/6-31 G(d,p) (Supplementary Structure 5)[30].

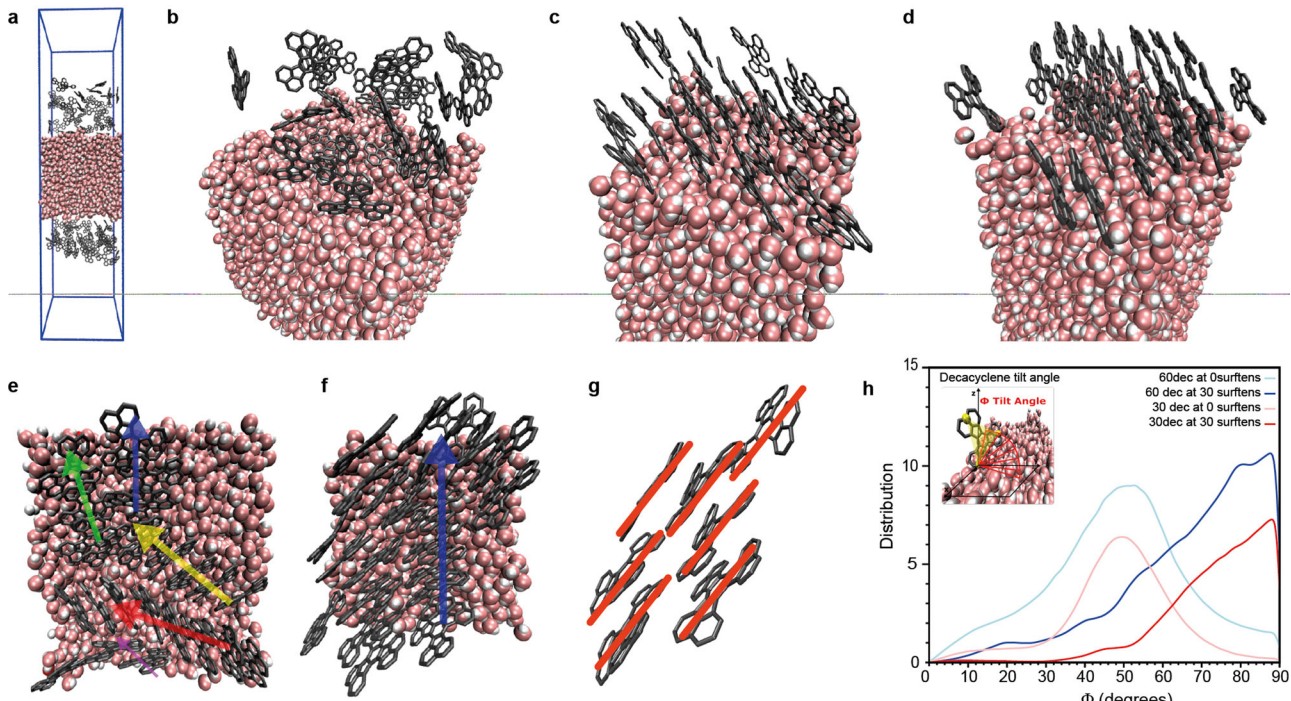

**Fig. 2 MD simulation on decacyclene. a** MD simulation box ($5.0 \times 5.0 \times 20.0$ nm$^3$) containing water molecules represented by a space filling model, and the random, pre-equilibration position of decacyclene molecules (30) on both sides thereof represented by a licorice model in gray. Protons explicitly drawn on water, but omitted from decacyclene. **b** Top view of **a**. **c** Same system as **b**, after equilibration at 300 K. **d** Same system as **c**, after compression to a surface pressure of 30 mN m$^{-1}$. **e** Top view of **c** after equilibration at 300 K. Differently colored arrows added to highlight different stacking domains. **f** Same system as in **d** after compression to a surface pressure of 30 mN m$^{-1}$. **g** Highlight of molecules found in **f** showing the onset of roof-tiling. **h** Distribution of the tilt angle ($\Phi$) per decacyclene molecule as derived from the MD simulations. The definition of the tilt angle ($\Phi$) as the arc between the plane of a decacyclene molecule (yellow) and the x–y plane of the water surface, is illustrated in the top left. Shown are the distributions of the tilt angles for systems consisting of 60 (blue) and 30 (red) decacyclene molecules, before (light) and after (dark) compression to a surface pressure of 30 mN m$^{-1}$. Distribution curves were obtained via Gaussian broadening with default standard deviation and normalized per amount of decacyclene molecules, using a Kernel Density Estimation to produce this plot with $N_{bins} = 1897$.

intermolecular distance between decacyclene molecules was found to be ~3.6 Å (Fig. 2c), which is in good agreement with that computed with DFT for a decacyclene dimer (3.68 Å), (Fig. 1c) and crystallographically (3.9 Å)[31]. The binding energy of the dimer was found to be 23.1 kcal mol$^{-1}$ in terms of Gibbs free energy at the PBE-D3(BJ)/6-31 G(d,p) level. In all cases, a parallel stacking mode was found most favorable. In the MD simulations, the stacked molecules were found to align themselves perpendicularly to the water surface, forming distinct domains with different relative orientations (Fig. 2d–f). Intuitively, such an arrangement decreases the total surface area of the hydrophobic, decacyclene molecules with the water surface. These findings thus served as an initial validation that our MD model is able to accurately describe the intermolecular interactions in this system.

After pre-equilibration of the system, the surface area of the simulation box was reduced in the x-y plane, corresponding to an increase in surface pressure in the Langmuir-Blodgett compression technique. Plotting of the surface pressure as a function of the MMA thus yields an isotherm, which can be compared to those obtained experimentally (Fig. 1b). Systems containing 30 and 60 decacyclene molecules were modeled. In the former case, a monolayer of molecules was found with an average MMA of approximately 44 Å$^2$/molecule; in the latter case, a bimolecular layer is obtained, with an MMA of roughly 26 Å$^2$/molecule. The computed isotherm for the bimolecular layer system was found to be in reasonable agreement with experimental data. In both cases, upon compression, molecules in their distinct domains were found to intercalate with neighboring domains to adopt a slipped stacking mode, a process called roof-tiling (Fig. 2g)[32]. When

using a racemic mixture of $D_3$ conformers, i.e. containing equal numbers of (+) and (−) rotating molecules, distinct domains are still formed, containing a random mixture of isomers. Such systems give isotherms identical to optically pure systems (Supplementary Fig. 13). Apart from domain reformations, we also observed a change in the orientation of the molecules to the water surface upon compression. Quantification of the tilt angle (i.e. the angle of the plane of a decacylene molecule to the surface of water), revealed that upon compression, the preferred tilt angle shifts from 50 to almost 90 degrees, driven by a desire to minimize the contact area with the water surface (Fig. 2h). For comparison, note that decacyclene was previously found to have an intrinsic preference to absorb parallel to the surface of copper(100) foil[33,34].

Thin Film Characterization. To characterize the thin films on a microscopic scale, samples were transferred onto a Si/SiO$_2$ wafer by horizontal lifting at constant pressure (25 mN m$^{-1}$) and imaged using atomic force microscopy (AFM) (Fig. 3a). This showed a uniform coverage of the substrate by the thin film, with a thickness of $2.5 \pm 0.7$ nm (Fig. 3b), corresponding roughly to a bimolecular layer, and corroborating the average thickness obtained from the MD simulations (vide supra). No crystallinity of the thin film was found with transmission electron microscopy (TEM) (Supplementary Fig. 4). To test the mechanical stability of the films, samples were transferred onto a copper TEM grid, covered with holey Quantifoil®[35]. Using scanning electron microscopy (SEM), films were found capable of being free-standing over 0.6 and 1.0 μm diameter apertures with only minimal defects (Fig. 3c). Transfer onto a 2 μm holey grid showed

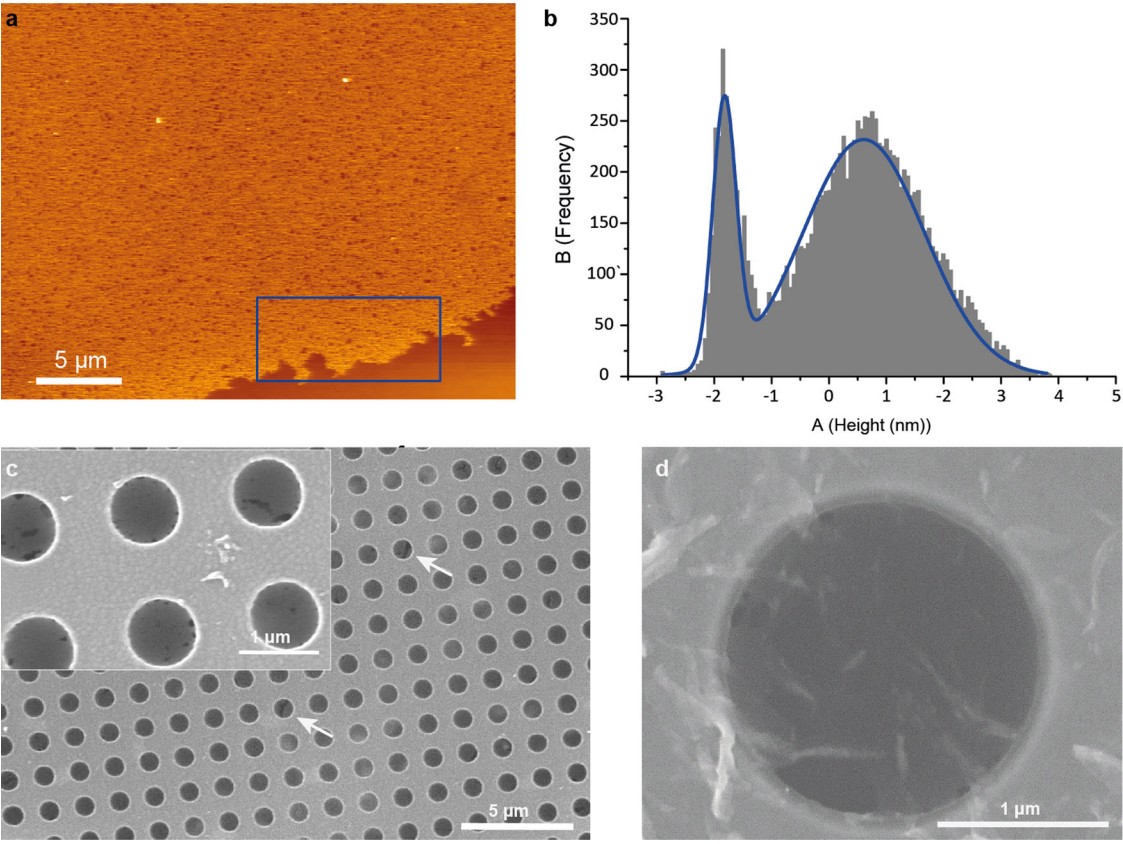

**Fig. 3 Thin film characterization. a** AFM image of a decacyclene film transfered unto a Si/SiO₂ wafer (For additional images see SI Fig. S5). **b** Height profile of the area marked in blue in **a**, showing a film thickness of 2.5 ± 0.7 nm. **c** SEM image of a decacyclene film free-standing over 0.6 μm diameter apertures. Ruptured thin films are indicated by arrows. Insert shows a zoom on the same grid, illustrating minor film defects as dark regions. **d** SEM image of a near-defect free decacyclene film free-standing over a 2 μm aperture.

films could be free-standing even over these larger distances (Fig. 3d), yet with less reproducibility.

Impressed by the ability of the thin films to be free-standing, without having required any form of annealing, we turned to quantify its mechanical strength. This was done using AFM nanoindentation experiments[36,37], using samples transferred onto a golden TEM grid covered with holey UltrAufoil®[38]. Initial images of the free-standing thin films showed a small height increase depicted as a lighter color around the perimeter of the aperture (Fig. 4a, Supplementary Fig. 6, Supplementary Fig. 7, ø = 1 μm). The thin films themselves were found to recede slightly into the apertures as a result of the Van der Waals attraction between the thin film and the TEM substrate (Fig. 4a, Supplementary Fig. 7). The same effect was previously observed for graphene[37,39], as well as for nanoparticle-based free-standing films, the latter of which are also held together by Van der Waals forces[40].

The freestanding thin films were indented in the center of the circular apertures with an AFM tip (nominal radius 2 nm) until failure of the decacyclene film occurred (Fig. 4a, b, Supplementary Fig. 8). For each indentation, the depth of the indentation by the AFM tip was plotted as a function of the applied force (Fig. 4c). The resulting force-indentation curves were analyzed using an elastic model for a 2D material in the form of a circular disk clamped along its circumference[37]. According to this model, the dependency of the applied force $F$ on the vertical tip position $\delta$ can be written as linear addition of pretension $\sigma^{2D}$ and the elastic modulus in 2D material $E^{2D}$:

$$F = \sigma^{2D}(\pi R)\left(\frac{\delta}{R}\right) + E^{2D}(q^3 R)\left(\frac{\delta}{R}\right)^3 \quad (1)$$

Here, $R$ is the radius of the clamped film and $q = \left(1.05 - 0.15\nu - 0.16\nu^2\right)^{-1}$ is a dimensionless constant depending on the Poisson's ratio of the material. Considering the forces holding the thin film together are dominantly of a Van der Waals character, a Poisson's ratio $\nu = 0.33$ was assumed, as previously used for freestanding nanoparticle membranes[40,41], and protein monolayers grown on water[42], which are all held together solely by Van der Waals interactions. The 2D approximation is only valid for systems where the film thickness $h$ is considerably smaller than the film diameter $R$, i.e. $R/h > 1$. In our case $R = 1$ μm and $h = 2.5$ nm, and thus the approximation holds.

After the first contact with the decacyclene film, the tip induces elastic deformation fitting to the theory (Fig. 4c, red curve). The fit is valid up to the point of non-elastic changes in the material, which appear as a change in the slope or a small drop in the force (Fig. 4c, red arrow; Supplementary Fig. 10). These non-elastic changes were previously observed for graphene flakes as well as a consequence of cracks induced in the thin film[43]. Such points are hence not considered as full rupture of the decacyclene film, which is here defined as an event when the applied force decreases considerably (>2 nN; Fig. 4c, green arrow). The distribution of penetration forces and the force inducing non-elastic deformations are shown in Fig. 4d, e (Supplementary Fig. 9a, b). On average, the thin films fails at an applied force of 26 ± 9 nN (N = 26) with cracks being induced at a force of 8 ± 4 nN (N = 27). Note that sharper AFM tips require smaller forces to induce failure of a material[37,44]. To objectively assess the material's properties, intrinsic parameters were needed.

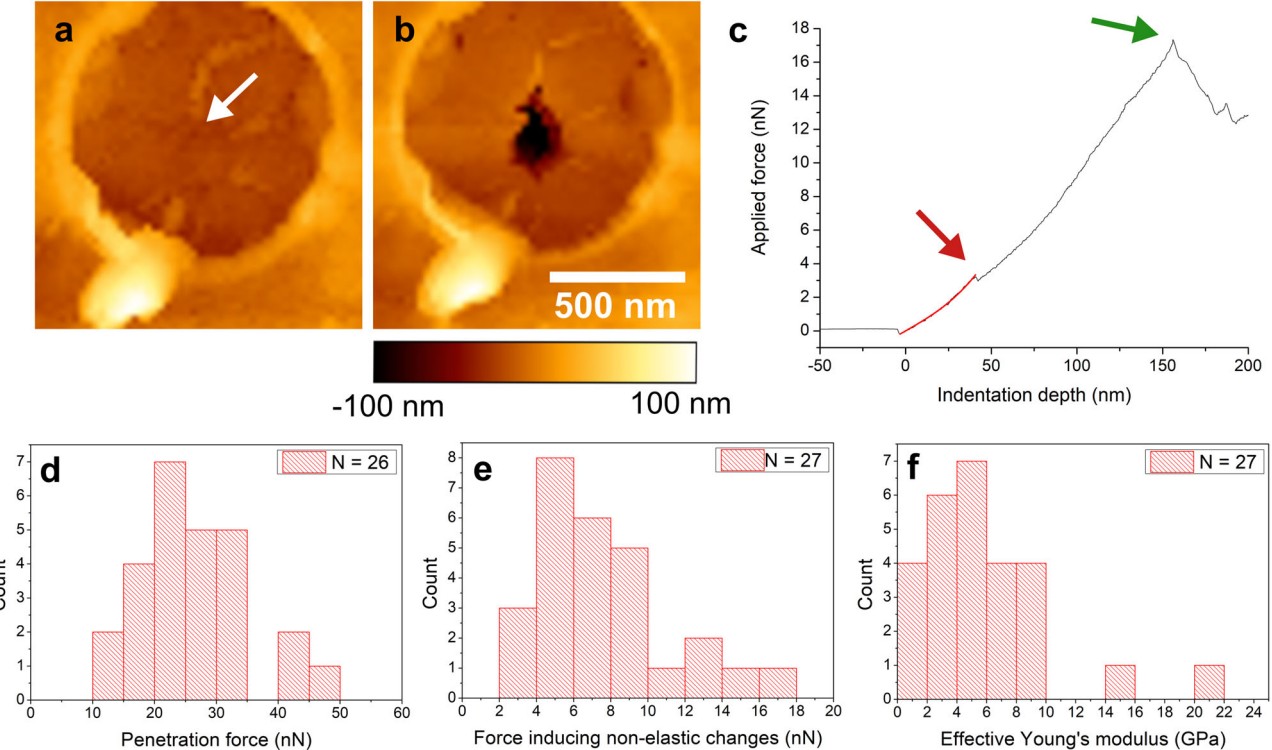

**Fig. 4 AFM nanoindentation of decacyclene films spanning over a 1 µm circular aperture.** Representative AFM images before **a** and after **b** nanoindentation. White arrow in **a** indicates the point of the indentation with the AFM tip. **c** Representative force-indentation curve. Red curve shows the fit in the elastic regime. Red arrow points to a small drop in the slope of the curve. Above this point the fit of the elastic properties is no longer valid. The green arrow shows the breaking event when the thin film fully ruptures. **d** Histogram of the penetration force needed to rupture the thin film. **e** Histogram of the force inducing non-elastic changes in the thin films. **f** Effective Young's modulus of decacyclene film.

The model described by Eq. 1 is a linear addition of the pretension and 2D elastic modulus. The fit approaches the prestress-independent limit when indented deep enough to produce nonlinear behavior. This condition is met here as, at deeper indentation, we obtain a successful fit that yields a 2D elastic modulus $E^{2D}$. We can derive the effective Young's modulus $E$ of the decacyclene film by dividing the 2D elastic modulus with the film thickness as $E = E^{2D}/h$. An histogram of the effective Young's moduli thus derived shows a narrow distribution with the mean value at 6 ± 4 GPa (N = 27) (Fig. 4f, Supplementary Fig. 9d). The resulting Young's modulus depends on the chosen Poisson's ratio of the material, which we choose based on literature ($v = 0.33$, vide supra). We also fitted the data with a Poisson's ratio for a graphene monolayer $v = 0.165$, in which case the Young's modulus values increase by ~14%, indicating only a moderate influence of Poisson's ratio on the results. The elastic strength of the presented decacyclene film is thus comparable with 2D MOFs (~5 GPa)[25], one order of magnitude less than 2D COFs (~26 GPa)[26], and three orders of magnitude less than pristine graphene (~1 TPa)[39].

To conclude, we here showed that the small propellerene decacyclene is able to form stable, free-standing films, held together solely by collective Van der Waals forces, and without the requirement of crosslinking or annealing. Computational modeling of the experimental conditions revealed the molecular reorientations and stacking interactions that take place at the air-water interface during Langmuir-Blodgett compression. Despite being held together by Van der Waals forces, films were found stable enough to be free-standing over micrometer distances, which allowed quantification of their mechanical strength using AFM nanoindentation experiments. Thin films are found capable of withstanding point loads of 26 ± 9 nN, with a corresponding Young's modules of 6 ± 4 GPa, which is on par with reported 2D membranes. Although the present thin films are not yet capable for use in a device setting, we believe these results constitute the starting point for further development of PAH-based, non-covalent thin films.

## Methods

**Sample preparation.** Decacyclene was synthesized and purified according to the procedure of Amick and Scott[27,45]. Identity and purity of the material were checked using ¹H NMR spectroscopy on a Bruker DPX 300 NMR instrument equipped with a BBFO probe head for 5 mm outer diameter tubes. Spectra were recorded at 300 MHz for ¹H using tetrachloroethane-$d_2$ obtained from a commercial source (SigmaAldrich) which was distilled before use. Solubility was found too low to allow recording of a ¹³C spectrum. NMR spectra were processed using the MestReNova© 14.1.0 software suite. FTIR spectra were recorded on a Perkin-Elmer Paragon 1000 FTIR spectrophotometer equipped with a Golden Gate attenuated total reflection (ATR) device. All spectra were found to be in accordance with literature. Melting points were recorded on a Stuart scientific SMP3 melting point apparatus and are uncorrected. Thin film samples were prepared on a Langmuir–Blodgett trough (KSV NIMA, Finland) filled with Millipore® Ultrapure water (18 MΩ cm⁻¹). In a typical experiment, decacyclene was dissolved in chloroform and filtered over a syringe filter (0.45 µm, PTFE, VWR™) to yield a clear brown solution. Concentrations were back-calculated from evaporated aliquots. Typical concentrations were on the order of 1 mM. A small amount of this solution (75 µL) was carefully spread at the air:water interface using a micropipette. After 30 min, the surface was compressed by two barriers with a fixed speed of 2 mm min⁻¹ to reach a certain surface. The surface pressure was measured with a Wilhelmy balance. Thin film samples were transferred onto silicon wafers at constant pressure by the Langmuir–Schäfer method. Samples were transferred onto copper or gold TEM grids following standard procedures[46]. Copper foil was obtained from Puratonic® and had a thickness of 0.025 mm and a 99.999% purity.

**Thin film characterization.** SEM images were recorded by using an FEI NOVA nano SEM 200 scanning electron microscope. Samples intended for indentation experiments were only imaged on a small part of the TEM grids, to prevent contamination by electron beam exposure. TEM experiments were conducted on an image-side Cs-corrected FEI Titan 80–300 microscope operated at 300 kV

(Supplementary Fig. 4). AFM images of the films on Si/SiO₂ wafer were recorded on a JPK Nano Wizard Ultra Speed machine with a silicon 254 probe (AC 160 TS, Asylum Research) with 300 kHz nominal resonance frequency (Supplementary Fig. 5). The images were scanned in intermittent contact mode in air at room temperature. Both the AFM images and the force curves were processed using JPK Data Processing software.

**AFM nanoindentation experiments**. AFM imaging and nanoindentation of the TEM grid supported freestanding thin films was performed with an JPK Nano Wizard Ultra Speed AFM[36,37,47]. The TEM grid (with apertures of 1 μm in diameter) was immobilized by tape on top of a glass slide and imaged in QI^TM mode. The experiments were performed in air at room temperature (22 °C) using SNL-10A cantilevers (Bruker) with a calibrated spring constant of $0.432 \pm 0.003 \, \text{Nm}^{-1}$ and a silicon nitride tip with nominal radius of 2 nm. The imaging force was ~400–600 pN. After imaging the free-standing films, the intact films were indented in the center by a force up to 50 nN with an indentation velocity of $300 \, \text{nms}^{-1}$. Force-distance curves were recorded during indentation. After indentation, the same spot was imaged again.

**Density functional theory**. Equilibrium geometries were computed at the PBE/6-31 G(d,p) level of theory using the Gaussian 09 Rev. D.01 program suite[48], using the D3(BJ) dispersion correction[49,50]. The geometry convergence criteria were set to tight (Opt = tight; Max. Force = $1.5 \cdot 10^{-7}$, Max. Displacement = $6.0 \cdot 10^{-7}$), and an internally defined super-fine grid size was used (SCF = tight, Int = VeryFineGrid), which is a pruned 175,974 grid for first-row atoms and a 250,974 grid for all other atoms. Free Gibbs energies were computed using Equation S2, in which $\Delta E_{\text{gas}}$ is the gas-phase energy (electronic energy) and $\Delta G_{\text{gas,QH}}^{T}$ ($T = 293.15$ K, $p = 1$ atm., $C = 1$ M) is the sum of corrections from the electronic energy to the free Gibbs energy in the quasi-harmonic oscillator approximation, including zero-point-vibrational energy. The $\Delta G_{\text{gas,QH}}^{T}$ were computed using the quasi-harmonic approximation in the gas phase according to the work of Truhlar in which vibrational frequencies lower than $100 \, \text{cm}^{-1}$ were raised to $100 \, \text{cm}^{-1}$ to correct for the breakdown of the harmonic oscillator model for the free energies of low-frequency vibrational modes[51,52]. Stationary points were checked to have no imaginary frequencies for local minima, and one imaginary frequency for the transition state structure. All DFT structures were illustrated using CYLview[53].

**Molecular dynamics**. All MD simulations were carried out using GROMACS 2016 software suite[54–60]. The Particle Mesh Ewald (PME) method was employed to accurately account for electrostatic interactions[61]. The cut-off for Coulomb and Lennard-Jones interactions was set to 10 Å. During the NVT simulation the temperature was kept fixed with the V-rescale coupling method[62].

The model for a body of water, comprised of 4139 water molecules, was simulated in a periodic box ($5.0 \times 5.0 \times 20.0 \, \text{nm}^3$) using the TIP4P-Ew/2004 force field[63]. The water surface tension was used as defined within the GROMACS software suite (Eq. 2):

$$\gamma_{water} = \frac{1}{2} L_z \left[ P_{zz} - \frac{1}{2} (P_{xx} + P_{yy}) \right] \tag{2}$$

where $L_z$ is the box length in the z direction, $P_{xx}$, $P_{yy}$ and $P_{zz}$ are the respective xx, yy and zz element of the pressure tensor[64], and the ½ originates from the presence of two x-y plane surfaces in the system. The system was first energetically minimized and then equilibrated for 8 ns with NVT at 300 K to obtain an average surface tension of $\gamma_{water} = 58.46 \, \text{mN m}^{-1}$. This value is in good agreement with previous studies[65,66]. The water model was further validated using a radial distribution function and density analysis (Supplementary Fig. 11). Decacyclene molecules were simulated using the OPLS-AA force field (Supplementary Table 1)[67–69] and LigParGen[70] parameterization with charges calculated with the CM5 model[71] using the Gaussian 16 Rev. C.01 program suite[72]. Input geometries for decacyclene were obtained from DFT as described above.

**Simulation run**. Random input positions of 30 or 60 decacyclene molecules on both water surfaces were generated using the PACKMOL18 program[73,74]. Having two independent water-decacyclene interfaces provides a more symmetric MD simulation box and increases the statistics of the results by averaging on both interfaces. Moreover, the presence of the decacyclene molecule on both sides avoids the diffusion of the water molecule through the periodic boundary condition along the z-axis. The two interfaces can be considered independent due to the thickness (5 nm) of the water box and the vacuum space (at least 5 nm on each side) above each interfaces along the z-axis. For each simulation, the system was first equilibrated with NVT at 70 K for 5 ns, and the temperature then raised to 300 K for another 5 ns. After these pre-equilibration simulations, the production run consists in NVT simulations were ran at 300 K for 10 ns. The final configuration extracted from the production run was used as a starting point for the subsequent NPT surface tension simulations. These series of simulations are repeated independently three times starting from a different random packing, in order to check the effect of initial conditions on the final results.

**Surface tension calculations**. Surface tension coupling for surfaces parallel to the xy-plane was used. Uses normal pressure coupling for the z-direction, while the surface tension is coupled to the x/y dimensions of the box. The surface pressure was then increased stepwise, to generate the pressure-area isotherm. At each chosen surface pressure (0, 3, 10, 20, 30, 40, 50 mN m⁻¹), the surface tension coupling molecular dynamics has been performed while monitoring the change of the area in the x-y plane. In order to execute the constant surface tension simulations, the Berendsen pressure coupling was used[75]. For this coupling methods to be effective, a value for the compressibility is required, which is close to the real compressibility of the system, namely $4.5 \times 10^{-5} \, \text{bar}^{-1}$[76]. Simulations were stopped after 10 ns at which point an equilibrium state was reached. Data presented in the manuscript was obtained averaging the values on the last 10 ns of each corresponding simulation. The x-y area values are equilibrated enough with an error around ±0.05 Å. (Supplementary Fig. 12). These series of simulations were repeated independently starting from each last equilibration simulation described in the previous paragraph, in order to increase the reproducibility of our results.

To verify whether a racemic mixture of decacyclene molecules gives the same isotherm as that of a homochiral system consisting of 60 (+) molecules, a system was modeled containing, above and below the water box, a mixture of 30 (+) rotating and 30 (−) molecules (Supplementary Structures 1, 2). No significant differences between the two systems was found (Supplementary Fig. 13).

All MD results were illustrated using Visual Molecular Dynamics (VMD)[77].

## Data availability
The authors declare that all data supporting the findings of this study are available in this article and Supplementary Information File, or are available from the corresponding author upon request.

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

## Acknowledgements

This research was supported with financial support of the European Research Council under the European Union's Seventh Framework Programme (FP/2007-2013)/ERC Grant Agreement no. 335879 project acronym "Biographene" and the Netherlands Organization for Scientific Research (NWO) VIDI 723.013.007, both awarded to G.S. A.M. was supported by an NWO Physics/f grant (no. 680-91-007). We acknowledge the Nederlandse Organisatie voor Wetenschappelijk Onderzoek (NWO) Exact and Natural Sciences for the use of the supercomputer facilities at SURFsara. The authors thank dr. Haoyuan Qi for performing the TEM characterization of the membrane.

## Author contributions

A.H. synthesized and characterized the compounds, performed the DFT computations, performed the Langmuir-Blodgett experiments, prepared the thin film samples and coordinated the work, X.L. helped with the Langmuir-Blodgett experiments and characterized the films by SEM and AFM. D.C. and M.K. performed and analyzed the MD simulations, A.M. performed the AFM indentation experiments on the suspended thin films, F.B. supervised and checked the computational aspects of the work, H.O. and D.F. supervised and checked the synthetic aspects of this work, W.R. and G.S. wrote the manuscript, interpreted the data and supervised the work. The authors acknowledge D.C. and A.M. contributed equally to this work.

## Competing interests

The authors declare no competing interests.
