## [Peer Review File · Nature Communications]

Freestanding Non-covalent Thin Films of the Propeller-shaped Polycyclic Aromatic Hydrocarbon DecacycleneREVIEWERS' COMMENTS

Reviewer #1 (Remarks to the Author):

The manuscript presents a method to prepare nanometer thin, porous membranes based on decacyclene. The experimental preparation is accompanied by some molecular dynamics studies and AFM nanomechanical characterization.

The authors have succeeded in preparing molecular thin films based on decacyclene and characterizing its mechanical strength on a freestanding configurations. Those achievements make this manuscript suitable for publication.

The manuscript does not report any specific applications for this type of membrane. In addition, other relevant experimental features such as the preparation of suspended membranes or the AFM method to measure the mechanical response of a manometer thin membrane were introduced by others. For those reasons, However, at this stage, I would recommend its publication in a more focused journal.

Reviewer #2 (Remarks to the Author):

- What are the noteworthy results?

Entirely 2D thin films based solely on non-covalent interactions between small molecules with backings from molecular dynamic simulations to understand conformational intermolecular interactions within the system. The elastic strength of the presented decacyclene membrane is thus comparable with 2D MOFs, held together solely by collective van der Waals forces, and without the requirement of crosslinking or annealing.

- Will the work be of significance to the field and related fields? How does it compare to the established literature? If the work is not original, please provide relevant references.

In terms of originality, the concept has been explored via graphene oxide-based systems, see referenced DOI: <https://doi.org/10.1016/j.chempr.2018.01.008>.

Though one can argue, that the authors of this paper have completed an in-depth work here that is significant to further push the notion that non-covalent systems are possible. Especially with the important data provided via MD simulations, unlike current literature studies which lack the studies that elucidate conformational/intermolecular interactions. On this basis, I feel that this article is strong and is important enough to be published.

- Does the work support the conclusions and claims, or is additional evidence needed?

Although the tensile strength of this material is particularly low in comparison to that of the referenced DOI, it is still interesting that there is a degree of film developed through the LB methodology outside what is found in literature, which is still comparable to the Young's Modulus of MOFs.

Are there any flaws in the data analysis, interpretation and conclusions? Do these prohibit publication or require revision?

The MD simulations have sufficiently backed claims of intermolecular interactions and agree with the hypothesis proposed during physical stress studies.

- Is the methodology sound? Does the work meet the expected standards in your field?

Yes.

- Is there enough detail provided in the methods for the work to be reproduced?

Yes.

Reviewer #3 (Remarks to the Author):

Interesting observation/fabrication of a free-standing stable film via pi-pi interactions. The abstract mentions "nanoporous" films, and "membranes" are also mentioned in the paper. But there is no evidence presented of any nanoporosity or any membrane behavior (transport of molecules, ions etc). If there are other areas in which such films could be important, experts in those areas should comment. As of now the paper appears - to my eyes - as being of very specialized interest.

REVIEWERS' COMMENTS

Reviewer #1 (Remarks to the Author):

The manuscript presents a method to prepare nanometer thin, porous membranes based on decacyclene. The experimental preparation is accompanied by some molecular dynamics studies and AFM nanomechanical characterization.

The authors have succeeded in preparing molecular thin films based on decacyclene and characterizing its mechanical strength on a freestanding configurations. Those achievements make this manuscript suitable for publication.

The manuscript does not report any specific applications for this type of membrane. In addition, other relevant experimental features such as the preparation of suspended membranes or the AFM method to measure the mechanical response of a nanometer thin membrane were introduced by others. For those reasons, However, at this stage, I would recommend its publication in a more focused journal.

Reviewer #2 (Remarks to the Author):

- What are the noteworthy results?

Entirely 2D thin films based solely on non-covalent interactions between small molecules with backings from molecular dynamic simulations to understand conformational intermolecular interactions within the system. The elastic strength of the presented decacyclene membrane is thus comparable with 2D MOFs, held together solely by collective van der Waals forces, and without the requirement of crosslinking or annealing.

- Will the work be of significance to the field and related fields? How does it compare to the established literature? If the work is not original, please provide relevant references.

In terms of originality, the concept has been explored via graphene oxide-based systems, see referenced DOI: <https://doi.org/10.1016/j.chempr.2018.01.008>.

Though one can argue, that the authors of this paper have completed an in-depth work here that is significant to further push the notion that non-covalent systems are possible. Especially with the important data provided via MD simulations, unlike current literature studies which lack the studies that elucidate conformational/intermolecular interactions. On this basis, I feel that this article is strong and is important enough to be published.

- Does the work support the conclusions and claims, or is additional evidence needed?

Although the tensile strength of this material is particularly low in comparison to that of the referenced DOI, it is still interesting that there is a degree of film developed through the LB methodology outside what is found in literature, which is still comparable to the Young's Modulus of MOFs.

Are there any flaws in the data analysis, interpretation and conclusions? Do these prohibit publication or require revision?

The MD simulations have sufficiently backed claims of intermolecular interactions and agree with the hypothesis proposed during physical stress studies.

- Is the methodology sound? Does the work meet the expected standards in your field?

Yes.

- Is there enough detail provided in the methods for the work to be reproduced?

Yes.

Reviewer #3 (Remarks to the Author):

Interesting observation/fabrication of a free-standing stable film via pi-pi interactions. The abstract mentions "nanoporous" films, and "membranes" are also mentioned in the paper. But there is no evidence presented of any nanoporosity or any membrane behavior (transport of molecules, ions etc). If there are other areas in which such films could be important, experts in those areas should comment. As of now the paper appears - to my eyes - as being of very specialized interest.

Response:

The authors thank all the reviewers for their careful reading of the manuscript. The authors thank reviewer #3 in particular for pointing out the discrepancy in use of the term "membrane(s)". The authors agree with the assessment of the reviewer that the term "membrane" should ideally only be used for those materials for which it is proven that they selectively let through certain materials. We have therefore replaced the term "membrane" by "thin film" or simply "film" where appropriate, in both the main text and supplementary information.